# Contrast-Enhanced Ultrasound in Cystic Endometrial Hyperplasia–Pyometra Complex in the Bitch: A Preliminary Study

**DOI:** 10.3390/ani10081368

**Published:** 2020-08-07

**Authors:** Marco Quartuccio, Luigi Liotta, Santo Cristarella, Giovanni Lanteri, Antonio Ieni, Tiziana D’Arrigo, Massimo De Majo

**Affiliations:** Department of Veterinary Science, University of Messina, 98168 Messina, Italy; luigi.liotta@unime.it (L.L); scristarella@unime.it (S.C.); glanteri@unime.it (G.L.); antonio.ieni@unime.it (A.I.); tizianadarrigo.vet@gmail.com (T.D.); mdemajo@unime.it (M.D.M.)

**Keywords:** dog, cystic endometrial hyperplasia, pyometra, contrast-enhanced ultrasonography

## Abstract

**Simple Summary:**

In canine cystic endometrial hyperplasia–pyometra (CEH) syndrome, toxic factors and endometrial inflammatory reactions are responsible of increased blood flow and lower vascular resistance in uterine arteries. Color Doppler ultrasound is regarded as an adjunctive tool for quantitative assessment of endometrial vascularization during uterine disorders. The aim of this study was to assess, through contrast-enhanced ultrasound (CEUS) exam, the vascularization in endometrial microvessels in CEH-pyometra in order to evaluate the possibility of application in this syndrome. In twelve female dogs with clinical symptoms related to pyometra, echographic, Color Doppler and CEUS exams were performed. Histopathological examination revealed severe CEH and pyometra, immunohistochemical stain with CD 34 confirmed the presence of angiogenesis. CEUS exam revealed a widespread, intense and rapidly developing homogeneous enhancement of the hyperplastic endometrium with absence of signal only in cystic areas. All parameters of the quantitative analysis were not significantly influenced by region of interest dimension and position. CEUS may improve not invasive evaluations in the CEH-pyometra syndrome and virtually in CEH-mucometra.

**Abstract:**

In cystic endometrial hyperplasia (CEH)–pyometra syndrome, toxic factors and endometrial remodeling culminate in changes characterized by exudative and degenerative inflammatory reaction. Recent studies on hemodynamic found an increased blood flow and lower vascular resistance in uterine arteries, suggesting color Doppler ultrasound as an adjunctive tool for quantitative assessment of endometrial vascularization during pyometra. The aim of this study was to assess, through contrast-enhanced ultrasound (CEUS) exam, the vascularization in endometrial microvessels in CEH-pyometra in order to evaluate the possibility of application in this syndrome. In twelve female dogs with clinical symptoms related to pyometra, B-mode, color Doppler and CEUS exams were performed. In CEH-pyometra uteri, histopathological examination revealed severe CEH and pyometra, immunohistochemical stain with CD 34 confirmed the presence of angiogenesis. CEUS exams revealed a widespread, intense and rapidly developing homogeneous enhancement of the hyperplastic endometrium, with absence of signal only in cystic areas. All parameters of the quantitative analysis were not significantly influenced by region of interest dimension and position. CEUS has the potential to improve clinical not invasive evaluations in the CEH-pyometra syndrome and virtually in CEH-mucometra.

## 1. Introduction

Cystic endometrial hyperplasia (CEH)–pyometra syndrome is a severe and frequent diestral disorder of intact bitches [1,2], its pathogenesis and differential diagnosis are a challenge to studies aimed at reproductive disorders. There are numerous responsible factors of the CEH–pyometra syndrome, such as the joint action of steroid hormones (progesterone and estrogen) during the different stages of estrus cycle; the insulin-like growth factor 1 (IGF-1) as responsible of endometrial proliferation; the bacteriological toxic factors, mainly due to *Escherichia coli*; the endometrial remodeling by matrix metalloproteinases until the endometrial changes characterized by exudative and degenerative inflammatory reaction [3]. Some authors hypothesize that differential diagnosis element between pyometra and diestrus or CEH-mucometra in bitches is marked by the simultaneous expression of inflammatory mediators and endometrium proliferative pattern [4,5,6,7]. In inflammatory disorders, angiogenesis is activated by VEGF (vascular endothelial growth factor), an angiogenic mediator in many cell types mainly related to cytokines such as interleukin-1-alpha and interleukin-6 [8,9]. VEGF acts through its FLT-1 and KDR receptors, which directly participate in the regulation of angiogenesis and induce cell differentiation and transport in the uterine lumen and glandular epithelium [10]. The activation of FLT-1 and KDR receptors by VEGF culminates in cell migration to the target tissue [11]. The evaluation of the inflammatory and endometrial responses with changes in uterine hemodynamic as an effect to pathologic changes could contribute to a better understanding of the mechanisms involved in the pathogenesis of this syndrome in dogs and, therefore, for a more precise diagnosis of the disease.

In dogs, ultrasonographic evaluation of the uterus permits determining the thickness of the uterine walls and the presence of luminal contents differentiating pregnancy from pathologic conditions [12,13]. In women and cows, the increase in blood flow due to uterine pathologic conditions can be detected by Doppler ultrasound, but in dogs, this technique has been used to evaluate the blood flow of the uterine arteries during estrus, normal and abnormal pregnancies and the puerperium [14,15,16,17].

Recently, a study on hemodynamic, inflammatory and proliferative mechanisms in the uterine tissue of bitches with CEH and Pyometra found an increased blood flow and lower vascular resistance, suggesting color Doppler ultrasound as an adjunctive tool for quantitative assessment of endometrial vascularization during pyometra [2].

In the last few years, contrast-enhanced ultrasound has been introduced to clinical practice in human and veterinary medicine. The research and advance in contrast-enhanced US (CEUS) imaging technology have been successfully applied in dogs in the evaluation of lesions of the liver, kidneys, spleen, urinary bladder, testes as well as prostate gland [18,19,20,21,22,23,24,25,26,27]. Studies conducted on rats in different stages of pregnancy and on macaques and humans have shown promising results in the dynamic evaluation of placental and intervillous perfusion [28,29,30,31], reducing worries concerning the use of contrast agents during pregnancy. Several authors have recently validated the CEUS method in mares and in bitches to evaluate the microperfusion of the uterus during pregnancy and vascularization of the mammary gland during the estrous cycle of dogs without any apparent side effects [32,33,34].

Literature data on CEUS applications in endometrial pathology are relatively few and concerning only human species [35,36].

The aim of this study was to assess, through CEUS exam, the vascularization in the microvessels due to endometrial angiogenic stimulus in CEH-pyometra in order to evaluate the possibility of application as a diagnostic and prognostic tool in this syndrome.

## 2. Materials and Methods

### 2.1. Ethical Approval

All treatments, housing and animal care were in compliance with EU Directive 2010/63/EU on the protection of animals used for scientific purposes and the experimental procedures following the current regulations (Official Gazette of the Region of Sicily, 6–4–2007, part I, n. 15, annex I). Informed consent was obtained from each dog owner before its inclusion in the study.

### 2.2. Animals and Study Design

This descriptive study included twelve privately owned bitches, nulliparous, 7 to 10 years old, 3 to 20 kg, of different genetic types: German Pinscher (3), Jack Russell Terrier (2) and mongrel dogs (7), 20–50 days after the end of the previous estrus cycle, identified on the basis of first appearance of the blood vulvar discharges, confirmed by vaginal cytology (parabasal and intermediate cells plus neutrophils) and serum progesterone levels (30–60 ng/mL), with clinical symptoms related to pyometra. The bitches were presented to the Veterinary Teaching Hospital of the University of Messina between February and September 2019.

Inclusion criteria were: no previous treatment with endogenous progestins or estrogens; clinical signs of systemic disease (fever, loss of appetite, depression, polydipsia, polyuria, vomiting, diarrhea and abdominal distension), leukocytosis (WBC > 18,000 cells/mL), moderate or intensive vulvar discharge purulent-like and, at the ultrasound examination, the presence of the cystic endometrial hyperplasia with uterine luminal content.

All dogs, after imaging procedures, were subjected to ovariohysterectomy and CEH-pyometra were confirmed by postoperative histopathological examination of the uterus.

### 2.3. B-Mode, Color Doppler and Contrast-Enhanced Ultrasound Procedures

Examinations with B-mode, color power Doppler ultrasonography and CEUS were performed on all dogs. Ultrasound examination was performed by the same investigator (MQ), a Mindray M9 ultrasound machine (Mindray Medical, Milan, Italy) equipped with a linear array broadband transducer (6.6 to 13.5 MHz) was used. Dogs were positioned in dorsal or lateral recumbency and gently restrained without sedation; abdominal wall was clipped, and acoustic gel was applied directly to the transducer. Standardized settings were used for depth (1–5 cm), overall gain, dynamic range and focal zone; time gain compensation was optimized in the near field.

B-mode exam of the uterus was performed to evaluate uterine lesions such as: dilation of the uterine horns, degree of thickening of the endometrium and size of endometrial cysts. The ovaries were examined to rule out the presence of other pathologies such as cysts, neoplasms, etc. [37]. Color power Doppler examination for detecting an increase of small vessels in the endometrial layer was used; a low pulsed repetition frequency (about 1 kHz) and a proper gain (60%), for detecting small vessels without artifacts, were set.

All CEUS exams were performed by the same two operators with more than 8 years of experience (MQ, MDM) using a linear (5.7 MHz) transducer with contrast agent capability. One person injected the contrast bolus and the other performed the ultrasonography exam. A low mechanical index (MI) from 0.05 to 0.07, persistency off, a wide dynamic range, and a single focal zone was set deeper to the endometrial lesion. The machine allowed a dual live function with B-mode and Contrast images displayed simultaneously. The ultrasound contrast medium used was sulfur hexafluoride (SonoVue^®^, Bracco imaging, Milan, Italy) and was prepared following the manufacturer’s recommendations; each vial of contrast agent (which contained 25 mg of freeze-dried powder) was reconstituted by injection of 5 mL of saline (0.9% NaCl) solution and shaken vigorously for 20 s. An aliquot (0.03 mL/kg of body weight) of the contrast medium was rapidly inoculated through a three-way valve connected to a 20 G intravenous catheter, inserted at the level of the cephalic vein, followed by 5 mL of physiological solution (0.9% NaCl), injected immediately after the contrast medium to wash the cannula and as a push in the venous circulation.

Two bolus injections were performed, the minimum interval between inoculations was at least 10 min and, between each injection, the residual bubbles were destroyed by a scan of the organ and aorta in fundamental mode with a power in 100% regulated output. The activation of the timer was performed simultaneously with the contrast agent dose inoculation. During the first bolus injection, if necessary, machine settings (i.e., total gain, time–gain compensation) were optimized and not changed anymore during the second injection, when a video of 180 s was recorded. The transducer was manually positioned by the same operator during each imaging procedure and the same picture, most representative of lesion, was maintained during the complete CEUS examination.

### 2.4. Imaging Analysis

Raw data obtained during the second contrast enhanced examination were stored and, subsequently, they were analyzed by the two operators (MQ, MDM).

Qualitative assessment of the contrast enhancement patterns during the wash-in and washout phases in endometrial hyperplasia provides information on the pattern, order and quantity of blood perfusion of the different layers of uterus.

Post processing quantitative analysis of video-clips was performed using the integrated specialized software contrast imaging quantitative analysis (QA) of the ultrasound machine (Mindray Medical, Milan, Italy), without movie clips exportation. Contrast imaging QA system adopts a time–intensity analysis to obtain perfusion quantification information of volume and velocity of flow. For each dog, 3 region of interests (ROIs) were manually drawn in the endometrium: two adjacent round ROIs (0.3 × 0.3 cm) located in a row and a larger ROI (0.8 × 0.5 cm) that encompassed the 2 smaller ROIs (Figure 1). The ROIs were located at a depth of approximately 1.0 to 2.0 cm within the endometrium in the near field. Considering the occurrence of very small interspersed endometrial cysts, ROIs were placed only avoiding larger ones. As possible, the ROIs locations were standardized among the dogs.

Quantitative parameters calculated include the following: GOF (goodness of fit): to calculate the fit degree of the curve; range: 0–1, where 1 means the fit curve fits the raw curve perfectly; BI (base intensity): basic intensity of no contrast agent perfusion status; AT (arrival time): time point where contrast intensity appears, generally, the actual time value is 110% higher than the base intensity; TTP (time to peak): time when the contrast intensity reaches peak value; PI (peak intensity): contrast peak intensity; AS (ascending slope): ascending slope of contrast, the slope between the start point of tissue perfusion to the peak; DT/2: time when the intensity is half the value of the peak intensity; DS (descending slope): descending slope of the curve; AUC (area under curve): to calculate the area under the time–intensity curves during contrast.

### 2.5. Sample Collection

Tissue samples were collected for histological examination and fixed in a 10% formalin solution for 12 h at room temperature. After re-washing in tap water, samples were rinsed in graded alcohol solutions and later cleared in xylene. Following paraffinization, tissue samples were embedded in paraffin wax at 56 °C, then 5 µm serial histological sections were obtained from a microtome. Sections were stained using hematoxylin–eosin (H&E). Immunohistochemical analysis was carried out on parallel xylene-coated slides. All sections were first treated in a moist chamber with 0.1% H_2_O_2_ in methanol to block the intrinsic peroxidase activity (30 min at RT); with normal sheep serum to prevent unspecific adherence of serum proteins; with monoclonal primary antibodies against CD34 (clone QBEnd10, wd 1:50); with sheep anti-mouse or anti-rabbit immunoglobulin antiserum (Behring Institute, Marburg, Germany; wd 1:25; 30 min at RT); and with mouse/rabbit anti-horseradish peroxidase–antiperoxidase complexes (DakoCytomation, Glostrup, Denmark; wd 1:25; 30 min at RT). For the demonstration of peroxidase activity, the sections were incubated in darkness for 10 min with 3-3′ diaminobenzidine tetrahydrochloride (Sigma Chemical, St. Louis, MO, USA), 100 mg of diaminobenzidine in 200 mL 0.03% hydrogen peroxide in phosphate-buffered saline (PBS). The nuclear counterstaining was performed using Mayer’s hemalum solution. In addition, human tissues such as cutaneous, mesothelial, glial, lymphatic, striated and cardiac muscular fragments were utilized as tissue positive controls for the abovementioned antisera. To test the specificity of each immunostaining in order to rule out the possibility of a nonspecific reaction, serial sections of each specimen were tested by replacing the specific antisera by either PBS or normal rabbit serum, thus obtaining negative controls.

### 2.6. Statistical Analysis

Data of quantitative parameters were analyzed by 1-way ANOVA [38], considering ROIs as variable. Separation of means was assessed by Tukey’s test, and differences were considered significant if *p* < 0.05. Results were reported as least squares means ± standard error of the mean.

## 3. Results

### 3.1. Ultrasonographic Examinations

Ultrasonographic examinations required about 30 min, CEUS procedure was well tolerated and no side effects were noted.

B-mode ultrasonographic exam of the uteri revealed an expansion of the uterine horns, presence of anechoic material with interspersed hyperechoic spots, irregular thickening, sometimes remarkable, with presence of polypoid exuberances of the endometrium that protruded into the uterine lumen. Anechoic cysts of different volumes (0.3 to 0.5 cm in diameter) were interspersed in the thickness of the mucosa (Figure 2, left). Ovaries showed the presence of small hypoechoic areas (attributable to corpora lutea) in both ovaries in all bitches, and no pathologic changes were suspected. The evaluation with color power Doppler allowed to highlight vascular signals mainly at the endometrial level (Figure 2, right). Power-Doppler investigation, when signals were detected, showed from mild to moderate endometrial vascularization. Blood flow signals were not seen in all images acquired in the same subject.

CEUS exams revealed a widespread and rapidly developing homogeneous enhancement of the hyperplastic endometrium at 6–8 s from inoculation of the contrast medium (wash-in); the wash-out showed a slow decrease of enhancement, with a persistence of echoes up to 2 min. The enhancement was particularly intense in the hyperplastic endometrium, with absence of signal only in cystic formation, but evidence of enhancement in very thin peripheral vessels or necrotic areas; in the thin muscle layer, only small perpendicular vessels were evident (Figure 3). Table 1 shows the results from ANOVA for the quantitative CEUS parameters recorded during the study. All parameters of the quantitative analysis were not significantly influenced by ROI areas.

### 3.2. Macroscopic Examination

On macroscopic examination, the uteri were increased in volume and asymmetry of the uterine horns. Upon opening the organs, it was possible to highlight the hyperplastic aspect of the endometrium, as well as the presence of ectasic endometrial cysts.

### 3.3. Histological Evaluation

Absence of ovarian pathologies was confirmed. The histopathological examination revealed several serious pictures of endometrial hyperplasia with the presence of numerous cysts variables in size and shape containing purulent material and presence of moderate edema, vascular congestion and focal hemorrhage. An increased endometrial thickness, number of endometrial glands, some of these ulcerated and necrotic and fibroblast proliferation were seen; moreover, different inflammatory cells (lymphocytes, macrophages, neutrophils and plasma cells) infiltrate the endometrial structure and myometrium (Figure 4). These lesions were compatible with groups III and IV of the classification of Dow [39] and severe CEH and hyperplastic pyometra according to the classification of De Bosschere [40]. Immunohistochemical stain with CD 34 showed a high positivity towards the vascular endothelium confirming the presence of angiogenesis (Figure 5).

## 4. Discussion

Although current recommendations and guidelines do not clearly identify CEUS indications in the gynecological area in human medicine, a recent review synthesizes current knowledge relating to CEUS applications in diagnosing endometrial disease in women. Possible practical applications in this field could relate to: (a) evidence of the vascular axis of small polyps; (b) evaluation of the degree of intramyometrial invasion in endometrial cancer; (c) alteration of the circulatory pattern in inflammatory lesions [34].

To the authors’ knowledge, this is the first application of contrast enhanced ultrasound in cystic endometrial hyperplasia–pyometra complex (CEH-P) of bitches. The widespread distribution of the contrast medium in the hyperplastic endometrial mucosa provides accurate qualitative and quantitative information about uterine microvascularization and attests severe vascularization. Previous studies, focusing on Doppler investigations of flow of uterine arteries, showed a high speed with low resistance flow (low resistance index, RI) in bitches with CEH-pyometra syndrome compared with normal subjects in diestrous and bitches with mucometra [2,17]. Authors related these hemodynamic results to endometrial neovascularization and increased tissue perfusion resulting from inflammation. In fact, the endometrial neovascularization in the pyometra group is confirmed by higher immunostaining of VEGF-A and its receptors (FLT-1 and KDR), and angiogenesis is associated with simultaneous increased uterine expression of the COX-2 inflammatory marker [2]. Differently, in another report, older bitches with endometrial hyperplasia showed RI values of uterine arteries higher than normal ones, in fact CEH increases with the age; Authors suspected uterine fibrosis as responsible for a limitation of the vasodilatory response [41]. Interestingly, Veiga et al. [2] made a qualitative evaluation with a subjective scoring of the endometrial vasculature that had higher score in the pyometra group, but they, despite indicating color Doppler ultrasound as an adjunctive clinical tool, did not propose any scheme for grading.

Contrast-enhanced results of the present study showed an avid, rapid enhancement of the endometrial proliferation of the uteri of all bitches that correlated with immunostaining with CD 34 in post-hysterectomy histological exam so, CEUS investigation accurately highlights the high tissue vascularization due to local angiogenesis demonstrated on immunohistochemical examination, differently from assessments of the endometrium of normal bitches in diestrous, highlighting only a mild spot enhancement suggesting a scarcity of vascularization (unpublished). The quantification of blood flow parameters was also reported, and statistical analysis between ROIs of different dimensions did not show significant results.

CEUS quantitative evaluations may be influenced by patient-related, contrast medium and technical variables, so standardization of methods is very important to minimize at least technical variability [42]. Future studies are warranted to investigate whether perfusion parameters may be useful in discriminating different uterine disorders.

Endometrial biopsy is the “gold standard” tool for obtaining information about uterine health in the bitches. Biopsy samples may be obtained transcervical or by abdominal laparoscopy or laparotomy [43,44], but biopsy specimens could be often too small, and can therefore easily suffer from artifacts [45]; furthermore, a high risk for uncontrolled hemorrhage has been reported for transcervical collection of biopsy specimens [46], as well as, an increased risk for secondary contamination of the uterus by vaginal biota [47]. These considerations on invasive procedures to obtain endometrial biopsy sample may lead us to consider other diagnostic tests, such as contrast-enhanced ultrasonography, with lesser impact on integrity and function of the uterus.

The main limitations of the present study are the small number of bitches and the criterion of inclusion of uteri with high grade of CEH-pyometra; evaluations in a larger samples of animals affected by CEH-pyometra, CEH-mucometra and diestrous females with no uterine pathologies are needed to evaluate the parameters obtained and to establish reference values in order to provide a more precise correlation between the ultrasound images obtained with the CEUS method and the histopathological reference frameworks provided by Dow and DeBosschere [39,40].

## 5. Conclusions

The potential usefulness of CEUS for the qualitative and quantitative characterization of uterine blood flow in CEH-pyometra featuring this method as a non-invasive and accurate tool for visualizing and quantifying endometrial perfusion modifications as response to pathologic changes. The technique offers significant advantages in terms of morphologic and functional evaluation of uterus together with patient safety.

We believe that clinical application of CEUS may contribute to the study of pathologic changes in the uterus and provide more precise indications on the choice of therapy (pharmacological or surgical) to be applied.

## Figures and Tables

**Figure 1 animals-10-01368-f001:**
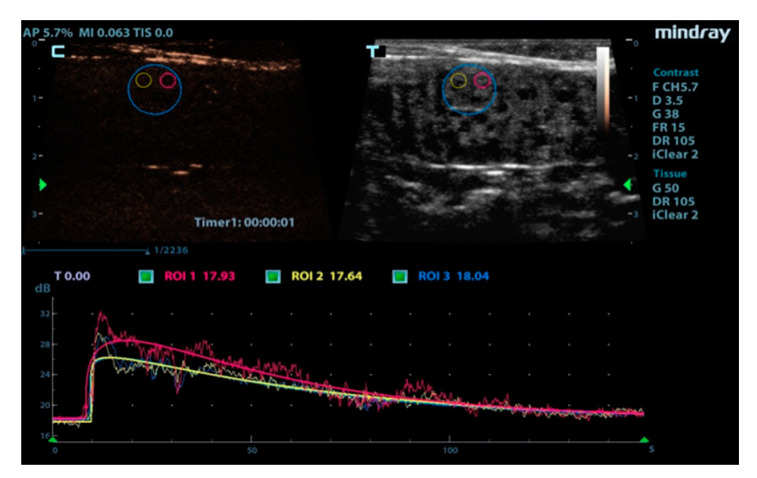
The region of interests (ROIs) were manually drawn into the near field of the endometrium: the large ROI encompassed the two smaller ones. At the bottom of the image raw curves (broken line) and fitted curves (continuous line) are consistent with the colors of the ROIs.

**Figure 2 animals-10-01368-f002:**
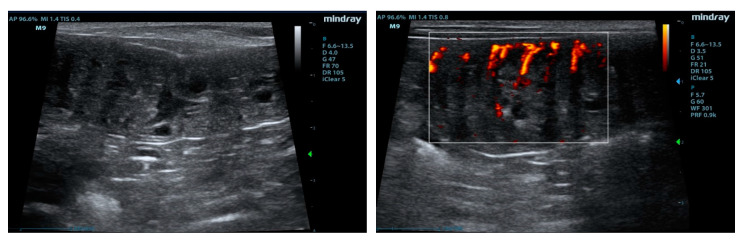
B-mode and color Doppler images of the uterus. Note the (**left**) cavities of cysts and (**right**) endometrial proliferation and vascular signals from the serosa toward the lumen.

**Figure 3 animals-10-01368-f003:**
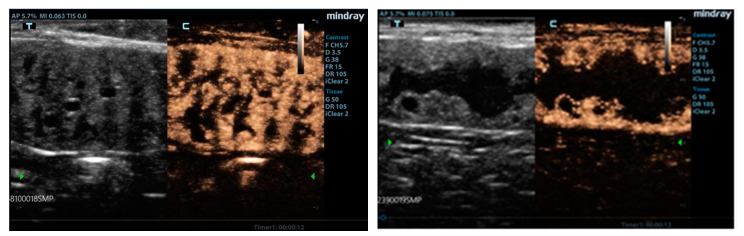
Contrast-enhanced images of the uterus. After 10–12 s hyperplastic endometrial layer showed (**left**) an avid enhancement with tortuous pattern due to presence of cysts and intraluminal material. (**right**) When uterine lumen is filled with material, very large cysts are surrounded by vascularized tissue.

**Figure 4 animals-10-01368-f004:**
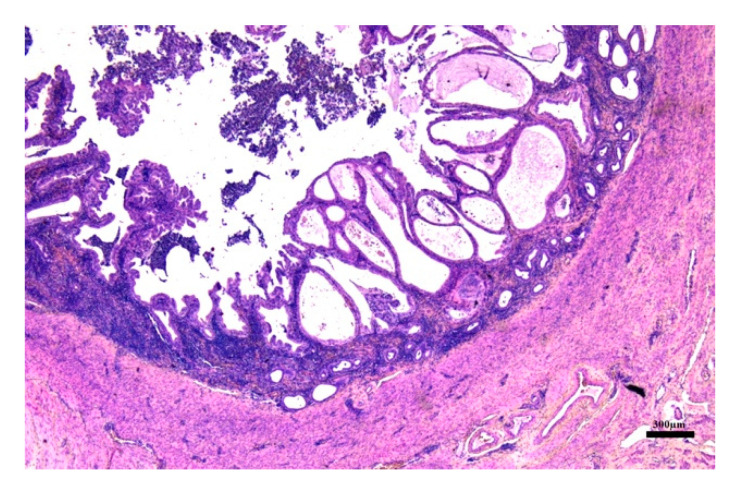
Uterus: endometrial hyperplasia–pyometra complex. Evidence of numerous cysts containing purulent material, vascular congestion and focal hemorrhage. Increased endometrial thickness and number of endometrial glands, some of these ulcerated and necrotic. (H&E, 2.5×).

**Figure 5 animals-10-01368-f005:**
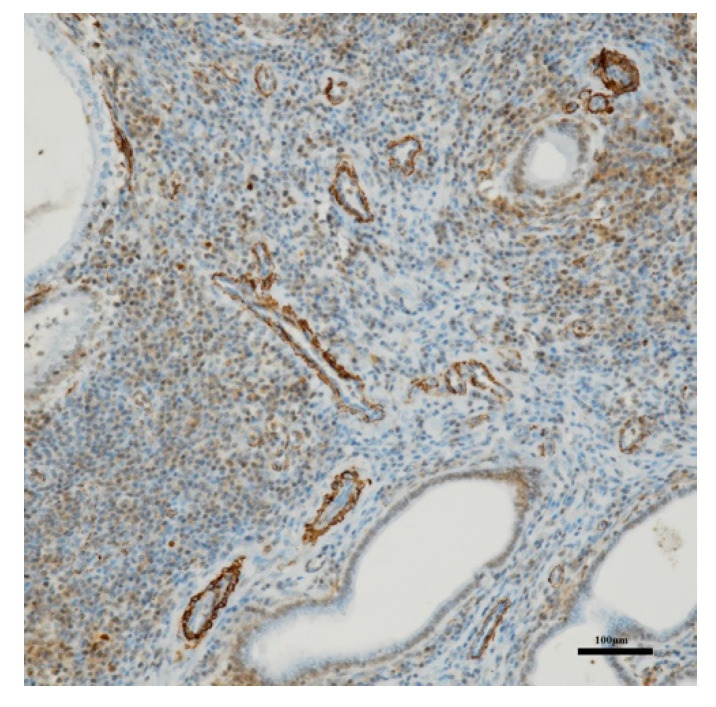
Uterus. Strong positive CD-34 reaction in the vascular endothelium. (10×).

**Table 1 animals-10-01368-t001:** Results from ANOVA for quantitative contrast-enhanced ultrasound (CEUS) parameters in relations to ROI* areas (mean values).

ROI Areas	No. of Animals	GOF	BI	AT	TTP	PI	AS	DT/2	DS	AUC
**ROI1**	12	0.93	18.94	0.13	6.93	26.46	0.41	47.26	−0.11	3034.04
**ROI2**	12	0.95	18.89	0.13	6.40	26.24	0.39	46.91	−0.08	3029.55
**ROI3**	12	0.95	18.11	0.12	6.62	26.22	0.46	47.74	−0.09	3078.44
**SEM**		0.05	0.45	0.01	0.33	0.54	0.06	0.55	0.01	11.08
***p*-value**		0.32	0.11	0.41	0.19	0.15	0.21	0.33	0.13	0.09

ROI—region of interests; GOF—goodness of fit; BI—base intensity; AT—arrival time; TTP—time to peak; PI—peak intensity; AS—ascending slope; DT/2—time when the intensity is half the value of the peak intensity; DS—descending slope; AUC—area under curve; SEM—standard error of the mean.

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
