# Peer review of "Contrast-Enhanced Ultrasound in Cystic Endometrial Hyperplasia–Pyometra Complex in the Bitch: A Preliminary Study"

_animals, 2020, doi:10.3390/ani10081368_

Round 1

Reviewer 1 Report

It a good experiment to use US contrast medium for diagnosis of pyometra.

CEUS is a well known technique, but its use in clinics is somewhat limited.   This paper has confirm the usefulness of

US contrast for the uterus-related disease.  But the main disadvantage is the high cost of US contrast, which

is a limitation fator to use US contrast widely in small animal clinics.

Generally it is well designed and written well. So I suggest it is worthy to be published in the journal, so that

many veterinarians could use CEUS in the reproductive system of small animals.

Author Response

Thanks you for positive comments on the paper submitted for a review.
In fact, the contrast medium has a relatively high cost, which, however, can be optimized in clinical practice by carrying out the investigation in two or more subjects consecutively.
To the knowledge of the Authors, the use of the CEUS investigation in the context of the bitch's uterine pathology is innovative.

Reviewer 2 Report

This article is a preliminary study about CEUS in cystic endometrial 2 hyperplasia-pyometra complex in the bitch. The manuscript is well-written.

I still about some comments and recommendations.

Ln 81: concerning the reproductive tract, CEUs of the normal mammary gland is also described in dogs. Please also include this in the introduction. (Anim Reprod Sci 2018 Dec;199:15-23. doi: 10.1016/j.anireprosci.2018.08.036. Epub 2018 Sep 13.B-mode and Contrast-Enhanced Ultrasonography of the Mammary Gland During the Estrous Cycle of Dogs Katrien Vanderperren 1, Jimmy H Saunders 2, Elke Van der Vekens 2, Eline Wydooghe 3, Hilde de Rooster 4, Luc Duchateau 5, Emmelie Stock 2 )

Ln 98-100: do you have any information about the dogs and the previous pregnancies? The authors mention that the dogs where 20-50 days after estrus. Was blood taken to confirm this?  

Lnn 119: what do you mean by type of the endometrial cysts?

Lnn 120: Concerning the color doppler, was this performed on the entire uterus and which parameters were used?

Lnn 136: please include in this section that a video of 180 sec was made. 2 injections were performed. The transducer was hold in the same place after the 10 min? You decided to perform CEUS on the part which was most affected? Can you please explanin more precise.

Ln 148: QA: please write full, what the authors mean without no data export?

Ln 150: ROI were placed within the endometrium on the side in the near field, so the endometrium closed to the transducer? Or also in the far field. Can you please clarify this.  I would also include a fig with the placement of the ROI. Where cystic lesions excluded from the ROI?

Ln 155: what is the goodness of fit?

Ln 195: mucosal-submucosal layers of the uterine wall, I would suggest the use the endometrium, myometrium and perimetrium. Please be consistent through the whole manuscript.

Ln 195: mucosa is used.

Ln 198: Ovaries showed the presence of corpus luteum in both ovaries in all bitches and the presence of pathologies was excluded. ‘And no pathologic changes were identified. ‘

Ln 120: Color Doppler; Lnn 199 Color Power Doppler was used. So both are different. Please change in the M&M To power doppler. The results second about power doppler is very short. The evaluation with Color Power-Doppler allowed to highlight vascular signals mainly at the endometrial level (Figure 1, right). Can you more specify the findings, mild, moderate, severe vascularization present? In the figure, you see mainly the doppler signal in the near field of the endometrium and not at the far field. Can you explain this?

Ln 262: I would include that CEH is increasing with the age. There are some articles about it.

276: ‘Future studies are warranted to investigate whether perfusion parameters might be useful in discriminating different uterine disorders’. What would the authors would like to differentiate with?

Reviewer 3 Report

Dear author,

Thank you for the interesting study. It opens the field of new possibilities in the study of CEH-pyometra complex using new imaging techniques.

However, some points seem unclear. It is important to insist on the fact that this is a descriptive study since no control group has been selected. In different parts of the manuscript, this point unclear or missing. It is clear that the use of CEUS is possible in the bitch but it is not possible to conclude that in your group of animals, CEUS shows an increased angiogenesis since you didn't compared it with healthy animals (even though the immunohistochemical analysis reflects it)

Here are the detailed corrections:

Line 102 to 106: this paragraph is incomplete. You should list all the possible clinical signs you observed. You diagnostic method of the pyometra is unclear. Did the bitches need all the criteria to be included? Leukocytosis is not observed in 100% of cases of pyometra so did you exclude bitches with no leukocytosis?

Line 105: "purulent uterine luminal content", you should change the formulation. You can't assess with a B-mode ultrasound exam that the uterine content is purulent, you only conclude it with your other clinical signs (for example purulent vaginal discharge)

Line 118: "presence or absence of exudates" this is unclear, the uterine horns can be dilated due to the absence of uterine exudates?

Line 122: unclear, all bitches were evaluated twice by two different operators?

Line 198: what are you criteria to define a corpus luteum with an ultrasound exam?

Line 199: how did you exclude ovarian pathologies?  A corpus luteum can be observed as an anechoic structure and ovarian cysts can have the same aspect. Other exams are needed to exclude ovarian pathologies that can sometimes be hard to diagnose with only an ultrasound exam

Line 254: I agree that CEUS here allows observing the microvascularisation. However, you can't observe a neoangiogenesis since there is no control groupe

Line 256: unclear

Line 268 to 273: here again it is not possible to conclude that CEUS is assessing an angiogenesis. CD34 does since it has been proven before with control groups but not CEUS. It seems to allow the same conclusion but a study with healthy bitch could confirm it.

Round 2

Reviewer 3 Report

Dear reviewer,

Thank you for your corrections. The fact that it is a descriptive study is now clearer. Your personal observations are interesting and I think that they could be added somehow in the discussion paragraph.

However, there are still some corrections to do.

The whole introduction has to be checked for spelling and grammatical corrections.

Line 46: an “are” must be missing

Line 47: “Numerous” not at the good place or sentence unclear

Line52 to 57: please rewrite those sentences, they are not understandable

Line 63: “responseS”, “Effect” and not “affect”

Line 66: “permitS”

Line 70: normal and abnormal pregnancIES

Line 80: replace “and” by “as well as”

Line 81: concerning the use of…

Line 83: microperfusion and not microperfmusion

Line 100-101: “on the basis of history appearance of the first blood vulvar discharge” unclear

Line 106: vomiting

Line 123: “in THE endometrial layer”

Line 125: “were set” and not setted

Line 144: “was recorded” instead of “was made”
